# CFD Analysis of Particle Dynamics in Accelerated Toroidal Systems for Enhanced PIVG Performance

**DOI:** 10.3390/mi15121432

**Published:** 2024-11-28

**Authors:** Ramy Elaswad, Naser El-Sheimy, Abdulmajeed Mohamad

**Affiliations:** 1Department of Mechanical and Manufacturing Engineering, Schulich School of Engineering, University of Calgary, 2500 University Dr. NW, Calgary, AB T2N 1N4, Canada; mohamad@ucalgary.ca; 2Department of Geomatics Engineering, Schulich School of Engineering, University of Calgary, 2500 University Dr. NW, Calgary, AB T2N 1N4, Canada; elsheimy@ucalgary.ca

**Keywords:** inertial navigation sensor, particle imaging velocimetry gyroscope, particle tracking, toroidal, computational fluid dynamics, discrete phase model

## Abstract

This study investigates the movements of particles in an accelerated toroidal flow channel filled with water, with specific applications for a particle imaging velocimetry gyroscope (PIVG). We used computational fluid dynamics (CFD) to simulate particle behavior under different angular accelerations. These angular accelerations were 4 rad/s^2^, 6 rad/s^2^, and 8 rad/s^2^ for particles densities of 1100 kg/m^3^, 1050 kg/m^3^, and 980 kg/m^3^. An examination was performed on the particles’ concentration distribution, velocity profiles, and displacement patterns with respect to the toroidal geometry, which had a volume fraction of 1.5% and was sized at 50 microns. Our results show that particle density significantly affects behavior and displacement within the toroidal flow, with heavier particles (1100 kg/m^3^) settling more quickly and concentrating near the lower z values over time, while lighter particles (980 kg/m^3^) maintain a more uniform distribution. This understanding is crucial for optimizing PIVG accuracy and reliability.

## 1. Introduction

A gyroscope is a device used for measuring or maintaining orientation and angular velocity. Gyroscopes, as rotation sensors, are part of inertial navigation sensors, which also include motion sensors (accelerometers). As inertial sensors’ accuracy has developed over time, they are now sufficient for navigation and guidance applications in terms of size, weight, cost, and accuracy. Fluid-based sensors as a class of inertial navigation sensors are very important due to their advantages of simple structures, low cost, high shock resistance, and long measurement ranges. However, their sensitivity and bandwidth are not comparable. In parallel, commercially available gyroscopes such as MEMS-based and optical gyroscopes have seen continuous evolution. MEMS gyroscopes are widely adopted for their cost-effectiveness and compact designs, making them ideal for consumer electronics and industrial applications. Meanwhile, fiber-optic gyroscopes (FOGs) and ring laser gyroscopes (RLGs) have emerged as leaders in high-precision environments, especially in aerospace and navigation systems. Recent improvements in these optical gyroscopes have enhanced their sensitivity and reduced noise, making them well suited for the high-precision measurement of angular velocity, further bridging the gap between fluid-based and more conventional gyroscopic technologies [1,2,3,4,5].

The particle imaging velocimetry gyroscope (PIVG) is a fluid-based sensor that was first introduced by Youssef A and El-Sheimy N [6]. A conceptual design for the PIVG is based on the particle imaging velocimetry technique which is an experimental tool of fluid dynamics science. The PIVG, as a single-axis sensor, consists of a fluid flow, flow channel, and imaging sensor, which is a digital camera, that is facing the flow channel to track a particle inside the channel which follows the actual dynamics of the flow, as shown in Figure 1a. The motion of such a particle is determined from the acquired sequence of images via a series of digital image processing techniques. Consequently, these images are used to specify the particle location. Hence, the location in each image is going to compare with the initial location to measure the particle’s velocity [6,7,8,9].

Modeling and simulation are critical to refine the conceptual design and optimize the technology of new sensors. In this work, we address the fluid-based design of the PIVG sensor. The methodology is using computational fluid dynamics (CFD) simulations—allowing us to probe into the behavior of fluids and particles inside the toroidal flow channel of the PIVG sensor illustrated in its coordinate system in Figure 1b. Moreover, the present study considers the relative tangential (RTV) of the fluid inside the toroidal channel and the behavior of solid particles with variations in the density and angular acceleration. This model would enable the interaction and distribution of the solid and fluid phases inside the toroidal channel to be well understood in detail and also benefit the performance as well as enhance the accuracy of PIVG [7,9].

Previous work by Elaswad, R. et al. [7,9] laid the foundation for optimizing the parameters of the PIVG by examining particle trajectories in accelerated toroidal systems using CFD-DPM simulations. Their research offered crucial insights into how particle behavior impacts gyroscopic performance, particularly under varying acceleration conditions. Furthermore, their analysis of the influence of the Dean number (D_e_) on flow dynamics within the toroidal structure highlighted the relationship between flow stability and sensor accuracy. These studies represent the first steps in the systematic development and optimization of PIVG parameters, which the current research builds upon by further exploring particle–fluid interactions and improving sensor performance [7,9].

Recent advancements in fluid dynamics within curved geometries and microchannels have deepened the understanding of complex flow behavior in systems pertinent to this research. Early foundational work by Dean, W. [10] examined the motion of fluid in curved pipes, introducing the concept of Dean vortices and stream-line motion, which significantly impact flow stability and particle distribution in curved environments. Building on this, Chen J. et al. [11] explored electroosmotic flow driven by both DC and AC electric fields in curved microchannels, highlighting how electric fields can induce unique flow patterns distinct from traditional pressure-driven flows. This finding is particularly relevant to toroidal structures like those analyzed in the present study, where curved geometries can dramatically influence fluid behavior. Furthermore, Chen K. et al. [12] investigated the effect of aspect ratio on laminar flow bifurcations in curved rectangular tubes driven by pressure gradients, illustrating the sensitivity of fluid dynamics to geometric parameters. Their work underscores how variations in geometry can impact flow stability and bifurcations, offering valuable insights into fluid dynamics in the PIVG sensor.

Research into particle dynamics in curved pipe flows has been extensively studied, with valuable insights provided by works such as the study by Henríquez Lira S et al. [13]. In their work on the numerical characterization of solid particle accumulation, they utilized Stokes numbers to quantify how curvature and turbulent flow regimes affect particle distribution and movement. This study highlighted the complex interplay between fluid dynamics and particle behavior in curved geometries, which is crucial for understanding flow stability and particle accumulation in various applications, including the PIVG sensor system. Additionally, the study by Noorani A et al. [14] on particle velocity and acceleration in turbulent bent pipe flows offers another layer of understanding by focusing on how turbulence in curved pipes influences particle velocities and accelerations. Their findings underscore the significant impact of flow curvature on particle behavior in turbulent environments, providing further context for the current work’s emphasis on particle tracking and distribution in the toroidal structure of the PIVG sensor.

These studies not only enhance the understanding of fluid dynamics in curved systems but also align with the objectives of the present work, which examines particle and fluid behavior within a toroidal system under varying conditions of angular velocity and particle density. By integrating these findings, this study aims to contribute to the broader discussion on complex flow patterns in complex geometries, while expanding the application of computational fluid dynamics (CFD) in analyzing gyroscopic systems.

## 2. Method

In the present study, a CFD simulation investigated how solid particles behaved in an accelerated toroidal chamber with a cross-sectional radius of 2.5 mm and a curvature radius of 25 mm. The basis of this simulation entailed the use of Navier–Stokes equations for fluid dynamics and the description of the conservation of mass and momentum, as in Equations (1) and (2) [7,15,16]. We defined the problem in cylindrical coordinates as illustrated in Figure 1a. The discrete phase model (DPM), utilizing a transient model, was employed to simulate the presence and behavior of solid particles within the toroidal environment. The discrete phase model (DPM), through a transient model, was activated to simulate the existence and dynamics of solid particles inside the toroidal domain. The particles were 50 microns in size and formed 1.5% volume fraction in the toroidal chamber. The particle injections were performed from 16 surfaces on the cross-section, as shown in Figure 1. Three different densities of the particles, namely 1100, 1050, and 980 kg/m^3^, were used, while the angular accelerations (*K*) imposed on the toroidal chamber were of the order of 4, 6, and 8 rad/s^2^.

All boundary conditions in the simulation were selected carefully to represent the physical system. The toroidal chamber, subjected to the no-slip condition as per Equation (4), is accelerated according to Equation (5), influencing the behavior of solid particles within the PIVG system. The no-slip boundary condition creates a velocity gradient at the interface, crucial for fluid dynamics and the behavior of particles. While the particles are approaching the boundary, the velocity difference between the bulk flow and the boundary creates shear forces, which then act to modify the formation of secondary flow structures. These forces affect the trajectory and distribution of particles inside this toroidal flow, especially when the densities of those particles differ. The no-slip condition thus helps in simulating realistic fluid–particle interactions within the toroidal chamber environment. The equations accounted for flow velocity (*v*), gravity (*g*), and body force from angular acceleration (*f*), and the fluid forces acting on particles are represented by $f_{pf}$. In our simulation setup, we modeled a transient, laminar flow regime using water (density $\rho_{f}=998.2\;kg/m$^3^, dynamic viscosity *μ* = 0.001003 kg/m·s). The walls were stationary, and the toroidal domain was subjected to rotating reference frame conditions with angular accelerated motion. Moreover, we used a hybrid initialization method with a time step size of 0.1 s and allowed up to 100 iterations per step, ensuring convergence with residuals limited to 1 × 10^−6^ for continuity and velocity.

The present work focuses on laminar flow and excludes turbulent flow. The exclusion of turbulent flow is performed for practical purposes that concern real applications of PIVG sensor. When the flow becomes turbulent, the velocity field becomes three-dimensional, with complicated vortices and velocity changes in all possible directions. This requires a complicated imaging system that captures 3-D velocity data of such a flow behavior, which is rather challenging for practical implementation. On the other hand, laminar flow permits us to consider the primary flow that is much easier to observe and measure in two dimensions. Thus, it would be practical to track the primary flow using a camera in real gyroscopic systems. In restricting the investigation to laminar flow, this serves to ensure that the experimental configuration is simplified for potential applications in which flow velocity monitoring using commonly available camera technology is more viable.
(1)$$\rho_{f}\left(\frac{\partial \vec{v}}{\partial t}+\vec{v}\cdot \nabla \vec{v}\right)=\rho_{f}g-\nabla p+\mu \nabla^{2}\vec{v}+\vec{f}+f_{pf}$$
(2)$$\vec{\nabla }\cdot \vec{v}=0$$
(3)v (at the wall)=0
(4)$$\omega (t)=\left\{\begin{array}{cc} 0 & for\;\;0\leq t<1\\ K(t-0.5) & for\;0.5<t\leq 4\\ 4\;K & for\;4<t\leq 8 \end{array}\right.$$

## 3. Results and Discussion

### 3.1. Grid Independence Study and Validation

A comprehensive grid refinement study was conducted to confirm the grid independence of the solution. Three meshes were created with a refinement ratio of approximately 1.5, as illustrated in Figure 2. The mesh refinement strategy involves categorizing the mesh into coarse, medium, and fine elements, with element sizes of (1.9309 × 10^−4^), (1.4586 × 10^−4^), and (1.0319 × 10^−4^) meters, respectively, and corresponding element counts of 427,680, 992,160, and 2,803,600. This study adheres to the guidelines recommended by Roache and Richardson [17,18]. The grid convergence index (GCI) method, as proposed by these authors, was employed to verify that the solution has achieved the asymptotic convergence range. The extrapolated values and GCI can be computed as follows for the meduim and fine elements [18]:(5)$$\underset{\;\mathrm{ext}(1,2)\;}{f}=\frac{\left(r^{p}f_{1}-f_{2}\right)}{\left(r_{1,2}^{P}-1\right)}$$
(6)$$(GCI{)}_{1,2}=\frac{1.25}{r^{p}-1}\frac{\left|f_{2}-f_{1}\right|}{f_{1}},$$

This is performed similarly for coarse and meduim elements, represented by $(GCI{)}_{2,3}$. Above, $f_{1}$ and $f_{2}$ are the solutions derived from the fine and coarse meshes, r is the refinement ratio, and p is the order of convergence.

During the validation stage, we will assess the accuracy and reliability of our numerical model by comparing the simulated fluid dynamics in the toroidal configuration with experimental data obtained from relevant studies. We also want to validate our particle distribution by performing tests and comparing our results with those obtained in a well-known lid-driven cavity experiment.

We compared our results with those of Madden et al. [19] in order to confirm our numerical simulation of a fluid-filled toroidal structure used in the PIVG sensor. Our numerical model replicated the dimensions and operating circumstances of the rotating water-filled torus used in their experiments. The angular velocity was raised using a nonlinear ramp, and for consistency, the simulations were run with a Reynolds number (Re) of 300. As can be seen in Figure 3a, our findings indicate that the solutions have converged because there has been no discernible change between them.

Due to the lack of experimental setup studies for particle motion in a toroidal structure, we compared our results with the flow of particles in a lid-driven cavity. Using particle image velocimetry (PIV) and particle tracking velocimetry (PTV) methodologies, the study in [20] examines the effects of seeding particle characteristics, including size, concentration, and surface coating, on flow attributes. Measurements of velocity fields in a lid-driven cavity experiment are the main focus of the study. The main conclusions show that particle size, particle concentration, and chemical treatment have the biggest effects on acceleration and velocity. This issue was replicated with similar settings in our numerical simulation. The study’s lid-driven chamber measured 80 cm in width and 80 cm in height, and its upper surface had a velocity of 0.26 m/s. Furthermore, as illustrated in Figure 3b, a thorough comparison of velocity profiles along the *x*-axis showed a high degree of agreement between simulation and experimental findings. This thorough agreement across all study components guarantees the accuracy and dependability of our numerical method, enhancing the validity of the results and their correspondence with experimental results.

### 3.2. CFD Results

In this study, we present the fluid dynamics and behavior of solid particles within the toroidal structure of the PIVG sensor, considering various densities and angular accelerations. A grid convergence study and validation have been conducted in prior work [7,9].

Figure 4 illustrates the relative tangential velocity (RTV) of the fluid at an angular acceleration of 4 rad/s^2^ at each time step. The velocity profiles exhibited a distinct parabolic shape, reaching their maximum at the center of the toroid. Notably, as the toroidal input acceleration increases, the fluid velocities follow it until the 4 s mark. Subsequently, there is a gradual decrease in velocities as the toroidal acceleration remains constant. To illustrate this behavior, we present Figure 4a, showcasing the velocity profiles from 0.5 to 4 s, and Figure 4b, which shows the profiles from 4.5 to 8 s. In fluid dynamics, the parabolic velocity profile is a characteristic signature of laminar flow in curved conduits. Fluid particles situated near the center line will experience less viscous drag than those nearer the walls, thus resulting in higher velocities at the core. This behavior is exactly what one expects as a consequence of the no-slip boundary condition at the wall, which indicates zero relative fluid velocity with respect to it. These findings highlight the crucial connection between observed velocity patterns and toroidal acceleration, emphasizing their significance for advancing the development of the PIVG sensor.

Figure 5a–c presents the relative tangential velocity (RTV) of the fluid and particles with densities of 980 kg/m^3^, 1050 kg/m^3^, and 1100 kg/m^3^ at angular accelerations of 4 rad/s^2^, 6 rad/s^2^, and 8 rad/s^2^ over an 8 s period. The velocities are calculated at the center of the cross-section of the toroidal structure. In all cases, the RTV increases sharply for both the fluid and particles, peaking at around 4 s. The fluid consistently exhibits the highest RTV, followed by the particles, with the lighter particles (980 kg/m^3^) achieving RTVs nearly equal to the fluid due to their lower mass, which results in less inertial resistance to changes in flow velocity, allowing for quicker acceleration in response to applied forces. In contrast, heavier particles (1050 kg/m^3^ and 1100 kg/m^3^) show progressively lower RTVs, as their increased inertia makes them less responsive to the same fluid dynamic forces, resulting in a lag in their velocity relative to the fluid. As the angular acceleration increases, the fluid’s RTV also increases, stabilizing at approximately 1.79 mm/s, 2.6 mm/s, and 3.5 mm/s for 4 rad/s^2^, 6 rad/s^2^, and 8 rad/s^2^, respectively. Meanwhile, the particles’ velocities stabilize at progressively lower values as their density increases, with the heaviest particles consistently exhibiting the lowest RTV in each case.

The displacement profiles of particles within the toroidal structure display a consistent exponential shape across all densities and angular accelerations, as shown in Figure 5d–f. At an angular acceleration of 4 rad/s^2^, the lower-density particles (980 kg/m^3^) reach a maximum displacement of approximately 12 mm, while the heavier particles (1050 kg/m^3^ and 1100 kg/m^3^) achieve displacements around 10 mm. As the angular acceleration increases to 6 rad/s^2^, the displacement for the lower-density particles rises to about 17.5 mm, compared to 16 mm for the heavier densities. At the highest acceleration of 8 rad/s^2^, the displacement peaks at approximately 23 mm for the lower-density particles, while the heavier particles reach around 22 mm. Regardless of density, all displacement profiles exhibit this exponential growth pattern, indicating that higher angular accelerations lead to increasingly rapid displacements. This behavior underscores the significant influence of both particle density and angular acceleration on the dynamics of displacement, providing essential insights for optimizing the design and performance of the PIVG sensor.

The distribution of particle concentration along the *z*-axis within the toroidal structure reveals distinct behaviors based on particle density and angular acceleration, as shown in Figure 6. For particles with a density of 1100 kg/m^3^ under an angular acceleration of 4 rad/s^2^, the concentration profile changes notably over time. Initially, at 0.5 s (Figure 6a), the particles exhibit a normal distribution along the *z*-axis. However, by 1 s (Figure 6b), a significant portion of the particles have settled near the bottom surface of the toroidal structure, indicating gravitational settling. This trend continues at 1.5 s (Figure 6c) and persists through 3 s (Figure 6d), with most particles concentrated at the bottom, illustrating the influence of gravitational forces on particle distribution. In contrast, for particles with a lower density of 980 kg/m^3^, the distribution along the *z*-axis remains relatively uniform and continuous from 0.5 s (Figure 6e) to 3 s (Figure 6h) under the same angular acceleration. This consistent distribution suggests that lighter particles experience less gravitational settling compared to their denser counterparts, highlighting the role of particle density in determining vertical concentration profiles within the toroidal environment.

Along with particle concentration, another crucial factor is particle dynamics, including their velocity relative to the fluid at varying densities and accelerations. The PIVG’s performance heavily depends on particles accurately mimicking the fluid’s motion. However, heavier particles (1100 kg/m^3^) show significant lag compared to lighter particles (980 kg/m^3^), which closely follow the fluid flow. The RTV results show that lighter particles achieve higher RTVs, nearly matching the fluid’s velocity, ensuring better tracking of angular motion. Heavier particles lag behind, which can result in measurement errors in the gyroscope’s output. As particle density increases, the measurement sensitivity decreases, highlighting the importance of particle selection based on the specific gyroscopic application.

The ability of the PIVG to measure angular velocity accurately relies on particles maintaining velocity profiles that match the fluid dynamics. Deviations from buoyancy-neutral conditions—as demonstrated by the varying densities of the particles—directly impact the measurement performance. The results from our simulations show that lighter particles provide more reliable measurements due to their closer alignment with fluid motion. Heavier particles, while providing useful concentration data, introduce lag in their velocity profiles, which could distort the angular velocity readings. This dynamic behavior suggests that for precise PIVG measurements, it is critical to use particles with densities close to the fluid. Deviations in particle density introduce noise and potential inaccuracies in velocity tracking, which can be detrimental to the gyroscope’s performance in applications requiring high precision.

Finaly, this present study represents the essential role that CFD plays in the analysis and optimization of operational efficiency in PIVGs. In this regard, fluid dynamics and particle interactions in the toroidal chamber are analyzed from a boarder perspective to develop light into selecting the characteristics of particles and parameters of flow that would enhance accuracy in measurement. These optimizations will directly affect the enhancement of the gyroscope’s capabilities for correct angular velocity detection—a very vital parameter in navigation systems. This CFD-based analysis minimizes errors, hence allowing for stable and predictable flow that provides the way for the development of a PIVG system with enhanced reliability and accuracy. Such improvements are of great importance in application fields that demand high levels of accuracy in navigation, such as autonomous vehicles, aerospace systems, and robotics.

## 4. Conclusions and Future Directions

This study conducts a detailed analysis of the fluid dynamics, particle behavior, and concentration distribution within the toroidal structure of the PIVG sensor. Through CFD simulations, we investigated the relative tangential velocities (RTVs) of both fluid and particles of varying densities, along with their displacement and distribution under different angular accelerations. The simulations demonstrate that lighter particles, such as those with a density of 980 kg/m^3^, closely follow the fluid’s velocity at all angular accelerations. At 4 s and at 4 rad/s^2^, both the fluid and the 980 kg/m^3^ particles maintain a velocity of 1.79 mm/s, with virtually no deviation. However, as particle density increases, the velocity decreases. The 1050 kg/m^3^ and 1100 kg/m^3^ particles exhibit velocities of 1.59 mm/s (11% lower) and 1.4 mm/s (21% lower), respectively, at the same angular acceleration and time step. This trend is consistent across all tested angular velocities, with the heaviest particles trailing behind the fluid and lighter particles by up to 13% and 11% at 6 and 8 rad/s^2^, respectively.

Furthermore, the distribution analysis shows that lighter particles achieve a more uniform dispersion within the flow, which is another factor crucial for maintaining sensor accuracy. Also, the displacement data illustrate how particle density influences the motion within the toroidal structure. At 4 rad/s^2^, the 980 kg/m^3^ particles achieve a total displacement of 11.9 mm, while the 1050 kg/m^3^ and 1100 kg/m^3^ particles move 10.6 mm (11% less) and 10.5 mm (12% less), respectively. As angular acceleration increases, the differences in displacement remain significant. At 8 rad/s^2^, the 980 kg/m^3^ particles reach a displacement of 23 mm, compared to 22 mm for the 1050 kg/m^3^ particles (4% less) and 21.9 mm for the 1100 kg/m^3^ particles (5% less).

The relations of particle density, velocity, and displacement demonstrate the importance of the selection of appropriate particle sizes in order to optimize the performance of the PIVG sensor. Lighter particles would more accurately track the motion of the fluid, which is critical for more precise gyroscopic measurements and better reliability of the navigation system. Future work will extend this analysis to more complex CFD models, validate results experimentally, and explore advanced materials for further sensor optimization, enhancing performance under varying environmental conditions.

## Figures and Tables

**Figure 1 micromachines-15-01432-f001:**
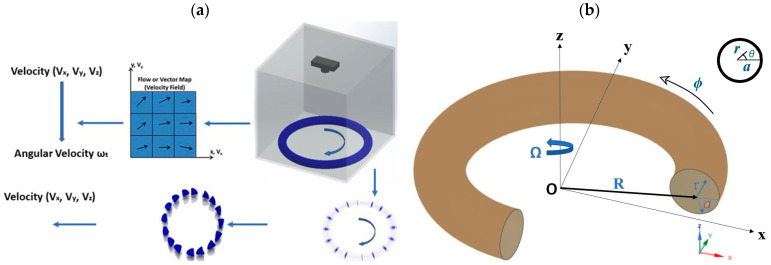
(**a**) The PIVG construction (single axis); (**b**) the toroidal coordinates.

**Figure 2 micromachines-15-01432-f002:**
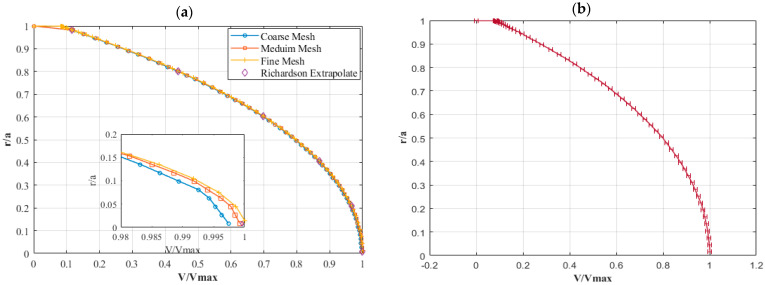
(**a**) The velocity profile functions of different No. of elements, at 2 s [7]. (**b**) Fine-grid solution, with discretization error bars computed using Equation (6).

**Figure 3 micromachines-15-01432-f003:**
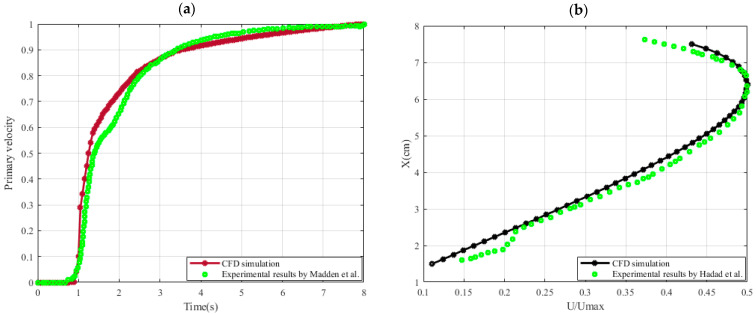
(**a**) Primary velocity vs. time series Madden et al. [19]. (**b**) The velocity profile on the horizontal line at the middle of the domain [7] and Hadad et al. [20].

**Figure 4 micromachines-15-01432-f004:**
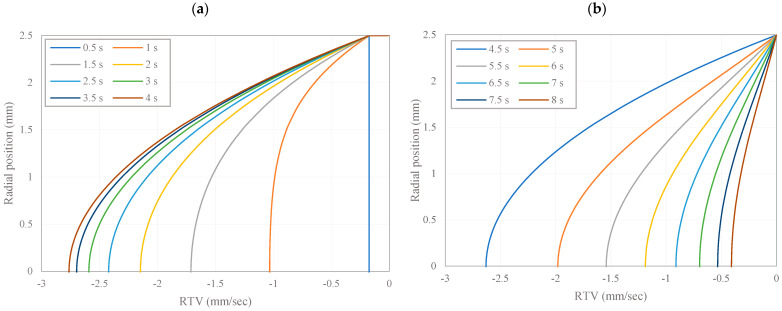
RTV profile for the fluid at different time steps at *K* = 4 rad/s^2^. (**a**) velocity profiles from 0.5 to 4 s, (**b**) profiles from 4.5 to 8 s.

**Figure 5 micromachines-15-01432-f005:**
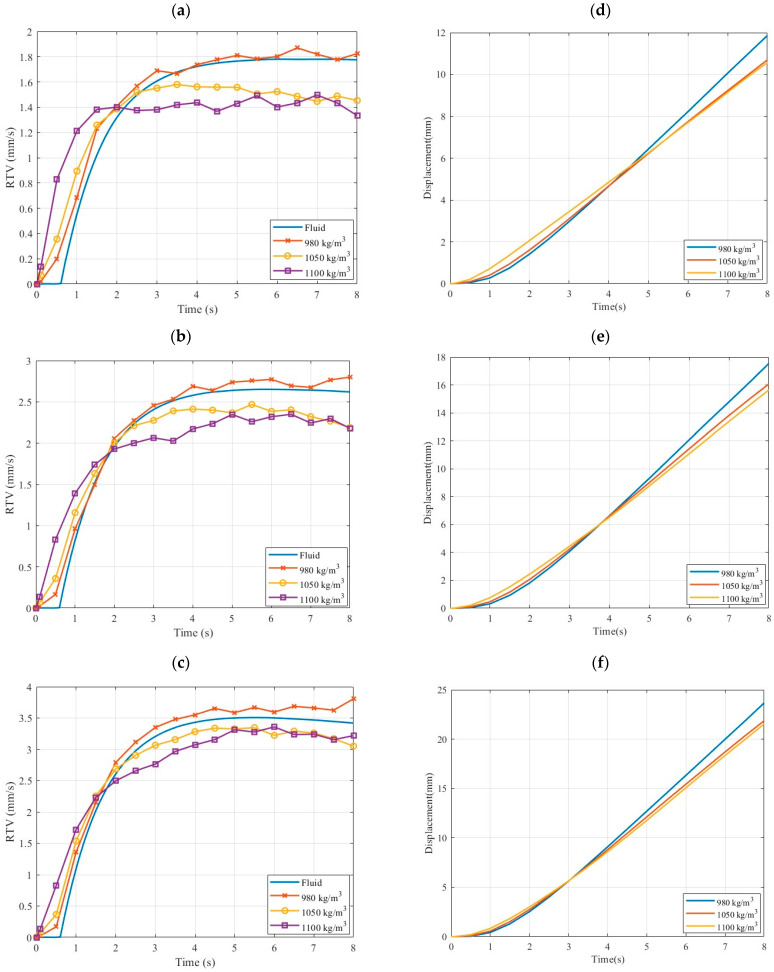
(**a**–**c**) RTV for fluid and particles at different densities and *K* = 4, 6, and 8 rad/s^2^, respectively. (**d**–**f**) Depiction of the displacement for the particles with different densities at *K* = 4, 6, and 8 rad/s^2^, respectively.

**Figure 6 micromachines-15-01432-f006:**
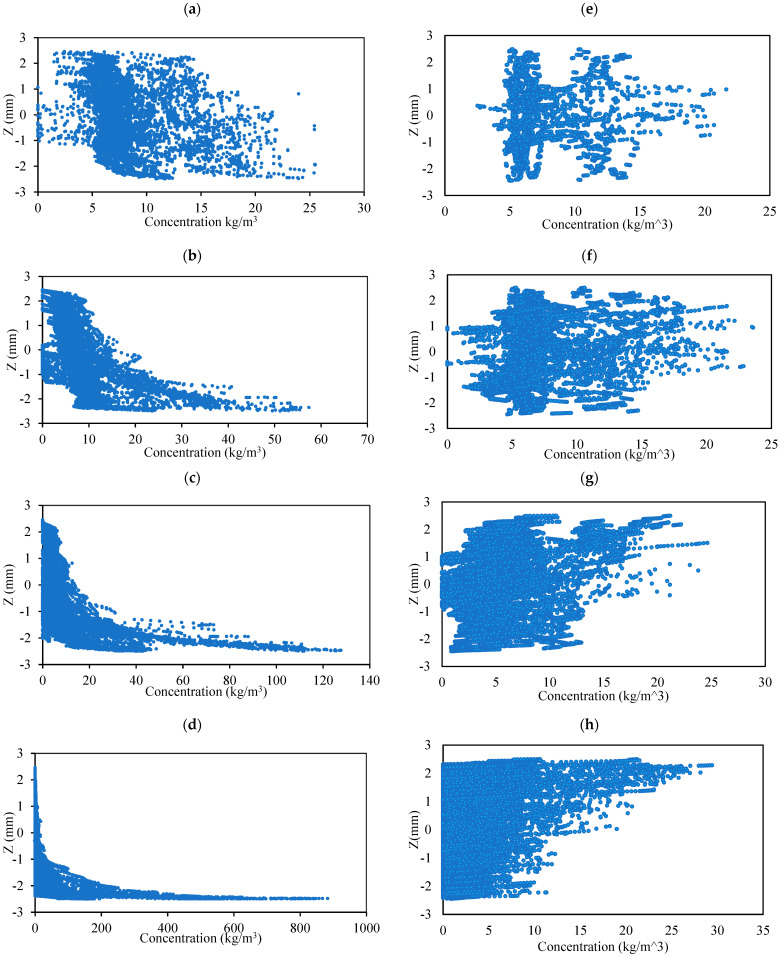
The concentration along the *z*-axis at different time intervals (0.5 s, 1 s, 1.5 s, and 3 s) for particles with densities of 1100 kg/m^3^ (**a**–**d**) and 980 kg/m^3^ (**e**–**h**), under an angular acceleration of *K* = 4 rad/s^2^.

## Data Availability

The original contributions presented in this study are included in the article. Further inquiries can be directed to the corresponding author.

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
