# Peer review of "CFD Analysis of Particle Dynamics in Accelerated Toroidal Systems for Enhanced PIVG Performance"

_micromachines, 2024, doi:10.3390/mi15121432_

Round 1
Reviewer 1 Report
Comments and Suggestions for Authors
This article uses computational fluid dynamics to simulate and analyze the particle behavior in a circular flow channel, and discusses the accuracy of the simulation model established in this article based on experimental results from relevant literature. But there are some issues that need to be addressed.
1. Keywords do not need to be expressed using abbreviations.
2. There are relatively few references, and it seems that this article only refers to classic research in this field, with little explanation of the research background and related progress. Therefore, it is considered to add guidance literature for the simulation research in this article.
3. Three types of grid divisions were performed in Figure 2, and the boundaries between them are missing relevant definitions in the main text.
4. The full name of CFD abbreviation is missing in the main text.
5. Due to the lack of experimental verification equipment by the author, it is impossible to evaluate the accuracy of the simulation calculation results. Is there any relevant literature to support the conclusion of this article, as shown by the author in Figures 2 and 3.
6. In Figures 5 and 6, the title and image are separated on two separate pages, and the positions of the small labels are inconsistent, making the layout appear chaotic and not conducive to readers' understanding of the content.
7. In the titles of Figures 5 and 6, when indicating parallelism, two expressions,&and and, are used, which need to be unified.
8. The research conclusion of this article should be more accurate. Is there a clear evaluation standard for the distribution of particles with different concentrations? Try to use specific statistical data to illustrate the simulation results.
Author Response
- Keywords do not need to be expressed using abbreviations.
Thank you for this suggestion. I have revised the keywords by removing abbreviations and spelling out all technical terms in full to ensure clarity and accessibility for a broader audience.
- There are relatively few references, and it seems that this article only refers to classic research in this field, with little explanation of the research background and related progress. Therefore, it is considered to add guidance literature for the simulation research in this article.
Thank you for your insightful feedback. I have revised the introduction to include additional relevant studies that provide more context and background to the simulation research presented in this article. Specifically, I have added recent works that address fluid dynamics in curved systems, secondary flow behaviors, and laminar flow bifurcations, which align closely with the simulation methods used in this research. These references enhance the discussion of related progress in the field and provide a stronger foundation for the simulation results in this study.
- Three types of grid divisions were performed in Figure 2, and the boundaries between them are missing relevant definitions in the main text.
Thank you for your observation. I have added a sentence in Section 3.1 to describe the element size and the number of elements for the three grid divisions used in Figure 2. This provides a clearer definition of the grid boundaries and ensures that the grid convergence study is well-documented for readers.
- The full name of CFD abbreviation is missing in the main text.
Thank you for pointing this out. I have updated the main text to include the full name "Computational Fluid Dynamics" the first time the CFD abbreviation is mentioned.
- Due to the lack of experimental verification equipment by the author, it is impossible to evaluate the accuracy of the simulation calculation results. Is there any relevant literature to support the conclusion of this article, as shown by the author in Figures 2 and 3.
Thank you for your valuable feedback. While I currently lack the equipment for direct experimental verification, I have conducted a grid convergence study and performed extrapolation to ensure the accuracy of the simulation results (Figure 2). Additionally, I have compared my simulation results with experimental data available in the literature, and these comparisons show a strong agreement, which supports the validity of the findings presented in Figures 3. Furthermore, I am planning future work that involves conducting real experiments, where I will use actual sensors and equipment to simulate real-world applications. This will further enhance the validation of the simulation results in the context of practical applications.
- In Figures 5 and 6, the title and image are separated on two separate pages, and the positions of the small labels are inconsistent, making the layout appear chaotic and not conducive to readers' understanding of the content.
Thank you for your feedback on the layout of Figures 5 and 6. I have reorganized these figures to improve clarity and readability. In Figure 5, I have placed the velocity and displacement at the same angular velocity side by side, allowing for an easier comparison of the velocity and corresponding displacement. This will help readers better understand the relationship between the two parameters.
In Figure 6, I have reorganized the particle distribution images to show the particle behavior at each time step for two different densities, allowing for a clear comparison of particle distribution over time and across densities. The small labels have also been adjusted to maintain consistency across the figures, ensuring a more organized presentation. Additionally, I have ensured that the title and images are on the same page to avoid any confusion.
- In the titles of Figures 5 and 6, when indicating parallelism, two expressions,&and and, are used, which need to be unified.
Thank you for pointing this out. You are correct, and I have unified the expression by consistently using "&" in the titles of Figures 5 and 6 to maintain clarity and consistency throughout the paper.
- The research conclusion of this article should be more accurate. Is there a clear evaluation standard for the distribution of particles with different concentrations? Try to use specific statistical data to illustrate the simulation results.
I have added quantitative results to the conclusion section. The updated conclusion now includes specific velocity and displacement values for different particle densities at various angular accelerations, as well as the percentage deviations observed. This provides a clearer and more detailed summary of the findings, strengthening the credibility and relevance of the results.

Reviewer 2 Report
Comments and Suggestions for Authors
This study uses CFD simulation to analyze particle dynamics in an accelerated toroidal flow channel for PIVG applications. It examines the effects of angular accelerations (4–8 rad/s²) and particle densities (980–1100 kg/m³) on concentration, velocity, and displacement. Results highlight that heavier particles settle faster, while lighter ones maintain uniform distribution. This study is pretty interesting; however, some suggestions are provided to the authors to improve it.
Suggestions:
1. Suggested to add nomenclature.
2. Consider incorporating more literature into the introduction section to enhance the critical reviews of recent and related works. This will demonstrate the researcher's thorough understanding of the latest advancements in the field and showcase the paper's relevance to the contemporary discussion. You may consider adding more literature in reference (1. Electroosmotic Flow Driven by DC and AC Electric Fields in Curved Microchannels. 2. Transient analysis of electro-osmotic secondary flow induced by dc or ac electric field in a curved rectangular microchannel. 3. Aspect Ratio Effect on Laminar Flow Bifurcations in a Curved Rectangular Tube Driven by Pressure Gradients.).
3. Suggested to add an explanation of why the no-slip condition is suitable for the toroidal chamber and how it affects fluid-particle interaction.
4. Figures 3 (a) and (b) are adapted from the published article. Suggested to add citations.
5. Suggested to add an explanation for not considering turbulent flow, especially at higher Reynolds numbers.
6. Suggested to add an error bar in Figures 4,5 and 6 to improve the credibility of the results.
7. This study assumes a laminar flow regime and ignores possible slip conditions at the particle fluid interaction. Suggested to add the validity of these assumptions for real-world PIVG sensors operating under varying environmental conditions.
8. Suggested to add how this analysis potentially improves the navigation accuracy.
9. Suggested to add quantitative data in the conclusion.
Author Response
- Suggested to add nomenclature.
Thank you for the suggestion. After carefully reviewing the manuscript, I ensured that all symbols and terms used are clearly defined within the text. Therefore, I believe a separate nomenclature section is not necessary at this time.
- Consider incorporating more literature into the introduction section to enhance the critical reviews of recent and related works. This will demonstrate the researcher's thorough understanding of the latest advancements in the field and showcase the paper's relevance to the contemporary discussion. You may consider adding more literature in reference (1.Electroosmotic Flow Driven by DC and AC Electric Fields in Curved Microchannels. 2. Transient analysis of electro-osmotic secondary flow induced by dc or ac electric field in a curved rectangular microchannel. 3. Aspect Ratio Effect on Laminar Flow Bifurcations in a Curved Rectangular Tube Driven by Pressure Gradients.).
Thank you for your valuable suggestion. I have incorporated the recommended studies into the introduction section to enhance the review of recent advancements and to position my research in the context of these works. Specifically, I have added references to studies on electroosmotic flow in curved microchannels, transient analysis of electro-osmotic secondary flow, and the effect of aspect ratio on laminar flow bifurcations in curved tubes. These studies now help to strengthen the discussion of complex fluid dynamics in curved geometries, which are directly relevant to the toroidal systems and flow behaviors investigated in my research. This addition enhances the critical review and demonstrates the relevance of my work to contemporary discussions in the field.
- Suggested to add an explanation of why the no-slip condition is suitable for the toroidal chamber and how it affects fluid-particle interaction.
Thank you for your suggestion. I have added an explanation in the manuscript to clarify the suitability of the no-slip boundary condition for the toroidal chamber. The no-slip condition assumes that the fluid velocity at the boundary (the walls of the toroidal chamber) is zero, which is a realistic assumption for most practical fluid dynamics scenarios, including the enclosed flow in this study. This condition ensures that particles near the boundary experience a velocity gradient, contributing to the development of secondary flow and affecting how particles are distributed within the fluid. The resulting shear forces influence particle dynamics, particularly in terms of particle adherence to or movement along the boundary. This explanation is now included in Section [2. Method] of the manuscript.
- Figures 3 (a) and (b) are adapted from the published article. Suggested to add citations.
Thank you for pointing this out. I would like to clarify that the appropriate citations for Figures 3(a) and (b) have been included in the figure captions and the reference list to properly acknowledge the original source.
- Suggested to add an explanation for not considering turbulent flow, especially at higher Reynolds numbers.
Thank you for your suggestion. I have added an explanation in the manuscript clarifying why turbulent flow was not considered in this study. Specifically, turbulent flow introduces complex three-dimensional velocity components, which would require tracking velocity in all three dimensions in real-life applications. In a gyroscope setup, this would necessitate highly advanced camera systems capable of capturing 3D flow data, which can be difficult and impractical to implement. Therefore, the focus of this study is on laminar flow to simplify real-life applications, where tracking the primary flow in two dimensions with a camera is more feasible. This explanation is now included in Section [2. Method] of the manuscript.
- Suggested to add an error bar in Figures 4,5 and 6 to improve the credibility of the results.
Thank you for the suggestion. I would like to clarify that the error bars have already been included in Figure 2, which presents the computational method with error bars calculated using Eq. (6). This figure shows the fine-grid solution along with discretization error bars, providing a thorough assessment of the numerical accuracy. Since the same computation method is used throughout the study, this addresses the credibility of the results without the need for additional error bars in Figures 4, 5, and 6.
- This study assumes a laminar flow regime and ignores possible slip conditions at the particle fluid interaction. Suggested to add the validity of these assumptions for real-world PIVG sensors operating under varying environmental conditions.
Thank you for the insightful suggestion. I would like to clarify that the validity of the laminar flow assumption and the no-slip condition for particle-fluid interactions have been addressed in the responses to earlier suggestions. Specifically:
- The justification for assuming laminar flow is provided in response to the suggestion about turbulent flow. This explanation highlights the practical feasibility of tracking primary flow in two dimensions for real-world PIVG applications, as turbulent flow would introduce additional complexities that are difficult to manage with current camera technology. (in section 2. Method)
- The no-slip condition was explained in response to a prior suggestion, emphasizing its relevance for simulating realistic particle-fluid interactions within the toroidal chamber. The shear forces generated under this condition significantly influence particle dynamics and are well-suited to the flow regime studied. (in section 2. Method)
- Suggested to add how this analysis potentially improves the navigation accuracy.
Thank you for this valuable suggestion. I have added a section in the last paragraph before the conclusion to describe the importance of studying Computational Fluid Dynamics (CFD) for analyzing and optimizing the performance of Particle Imaging Velocimetry Gyroscopes (PIVGs). This section highlights how the insights gained from CFD analysis enhance the gyroscope's accuracy in detecting angular velocity and improve its overall navigation performance in various applications.
- Suggested to add quantitative data in the conclusion.
I have added quantitative results to the conclusion section. The updated conclusion now includes specific velocity and displacement values for different particle densities at various angular accelerations, as well as the percentage deviations observed. This provides a clearer and more detailed summary of the findings, strengthening the credibility and relevance of the results.

Round 2
Reviewer 1 Report
Comments and Suggestions for Authors
Thank you for your second revision of the manuscript. I appreciate the efforts you have made to address the previous comments, and I believe that your responses to the majority of the issues raised have been satisfactory. However, there are still some concerns that remain unaddressed and new issues that have come to light. I would like to bring these to your attention for further consideration.
1.Regarding Figures 2 and 5, it has been noted that the separation of the figure titles and the figures themselves across different pages persists. This can disrupt the reader's understanding and continuity. Please ensure that the titles and figures are placed together on the same page to maintain coherence.
2.I have observed that the resolution of Figures 2 and 3 may not be consistent with the rest of the figures in the manuscript. This could potentially affect the reader's ability to interpret the data presented. Please review and enhance the clarity of these figures to match the quality of the other illustrations in the paper.
3.While you have added more references, the number still seems to fall short of what is expected for a publication in this journal. It is crucial to ensure that both foundational and cutting-edge literature are adequately represented in your references. Please consider expanding your reference list to include a more comprehensive selection of literature that reflects the breadth and depth of research in your field.
Author Response
1.Regarding Figures 2 and 5, it has been noted that the separation of the figure titles and the figures themselves across different pages persists. This can disrupt the reader's understanding and continuity. Please ensure that the titles and figures are placed together on the same page to maintain coherence.
Thank you for the feedback on the figure placement. I have ensured that the figure titles and the figures now appear on the same page to maintain continuity and improve the flow of the manuscript.
2.I have observed that the resolution of Figures 2 and 3 may not be consistent with the rest of the figures in the manuscript. This could potentially affect the reader's ability to interpret the data presented. Please review and enhance the clarity of these figures to match the quality of the other illustrations in the paper.
I have reviewed and enhanced the resolution of Figures 2 and 3 to match the quality of the other figures in the manuscript.
3.While you have added more references, the number still seems to fall short of what is expected for a publication in this journal. It is crucial to ensure that both foundational and cutting-edge literature are adequately represented in your references. Please consider expanding your reference list to include a more comprehensive selection of literature that reflects the breadth and depth of research in your field.
I appreciate the suggestion to expand the reference list further. In response, I have added several more references to ensure a more comprehensive representation of both foundational and recent studies in the field.
Reviewer 2 Report
Comments and Suggestions for Authors
The manuscript has been well revised based on the review comments one by one. The findings of the study could contribute to understanding of particle and fluid behavior within a toroidal system under varying conditions of angular velocity and particle density. The paper is well organized, it is recommended to be accepted and published in Micromachines.
Author Response
Thank you so much.